# Efficient Multi-Adapter LLM Serving via Cross-Model KV-Cache Reuse with Activated LoRA

## Abstract

Modern large language model (LLM) systems increasingly rely on multi-turn pipelines that are composed of multiple task-specific adapters, yet existing serving frameworks remain inefficient, incurring substantial recomputation overhead when switching between adapters. We present the first LLM serving engine that supports cross-model prefix cache reuse between base and adapted models via Activated LoRA (aLoRA), enabling efficient and fine-grained adapter switching during inference. Our design extends the vLLM framework by introducing base-aligned block hashing and activation-aware masking within the model execution path, permitting cache reuse across models while preserving compatibility with existing serving engine optimizations. Integrated into a production-grade inference stack, this approach supports dynamic adapter activation without excessive key-value tensor recomputation. Evaluation across representative multi-turn, multi-adapter pipelines demonstrates up to $58\times$ end-to-end latency reduction and over $100\times$ time-to-first-token improvement relative to standard LoRA baselines, with benefits that scale with model size and sequence length and manifest across all stages of the request lifecycle. This work bridges parameter-efficient model adaptation with high-performance serving, providing the first complete realization of cross-model KV-cache reuse in modern LLM inference engines.

## 1 Introduction

In recent years, the rise of large language models (LLMs) has spurred growing demand for model specialization, a trait essential for LLM adoption in narrow vertical markets requiring extensive domain-specific knowledge. LLM fine-tuning enhances pretrained LLMs by adapting their knowledge to a specific domain or task without compromising the model's core language capabilities. In contrast to full-finetuning methods, which face obstacles in computational resource constraints, long training times, overfitting, and potentially "catastrophic forgetting" (Nobari et al., 2025), parameter-efficient fine-tuning (PEFT) methods such as low-rank adaptation (LoRA) adapters freeze most of the foundation model's weights during training and adjust only rank $r$ additive adapters to target weight matrices. LoRAs have gained widespread adoption in practice due to their low resource requirements, often comparable performance to full finetuning, and the inference-time modular flexibility that they offer since adapters can be easily and efficiently switched in and out (Hu et al., 2021) on a single instance of the served base model.

As LLMs are increasingly deployed in complex reasoning and agentic pipelines (OpenAI et al., 2024; de Lamo Castrillo et al., 2025; Yao et al., 2023; Song et al., 2023), inference workloads are no longer dominated by single-task models. Instead, modern AI systems orchestrate multiple components—each responsible for carrying out specialized tasks such as safety checking and prompt rewriting, or invoking external tools such as APIs—within long multi-turn interactions (Zeng et al., 2025; Feng et al., 2025; Chen et al., 2025; Jin et al., 2025). This multi-adapter composition allows systems to leverage finetuned expertise dynamically during inference. While current serving frameworks are able to easily serve multiple LoRA adapters in heterogeneous batches, they still incur substantial overhead when switching adapters mid-sequence: every adapter change invalidates the model's key-value (KV) cache and forces a full recomputation of context representations before generation can resume. This cache invalidation problem becomes a bottleneck in

multi-turn or long-context pipelines, where the same input tokens may be repeatedly re-encoded for different adapters.

Activated LoRA (aLoRA) (Greenewald et al., 2025) introduces a mechanism to mitigate this inefficiency by modifying the model's attention projections only after a predefined activation sequence. Because the pre-activation attention weights remain identical between the base model and the adapter, the base model's KV-cache can be reused up to the activation point. This property makes aLoRA well-suited for pipelines where multiple lightweight adapters need to be invoked mid-generation with high frequency. In principle, aLoRA enables low-latency switching between model adapters, but realizing such fine-grained cache reuse in modern LLM serving systems presents nontrivial design challenges.

In this work, we design, implement, and evaluate a serving architecture that enables cross-model KV-cache reuse between base and aLoRA adapters, allowing models to toggle between specialized adapter modules during inference without redundant recomputation. Our system extends vLLM's existing prefix caching mechanism by aligning the hashing semantics of base and aLoRA-prefilled cache blocks, and expands the model execution pipeline to support aLoRA adapters through activation-aware masking into the model forward path. By enabling efficient, fine-grained adapter switching, our system bridges the gap between model-level specialization and system-level efficiency—a critical step towards deployable and dynamic LLM services.

Our contributions are as follows:

- We provide the first serving engine that supports Activated LoRA, modifying the vLLM framework and enabling efficient and frequent adapter switching in multi-turn inference via activated LoRAs and KV-cache reuse. In particular, enabled by the Activated LoRA architecture, we provide the first[1] state-of-the-art serving engine that supports prefix cache reuse *between* (adapted) models, rather than simply between consecutive calls to the *same* model.

- We evaluate the performance of our implementation in a set of illustrative multi-turn pipelines, realizing speedups from the cross-model prefix cache reuse, where the baseline is considered to be traditional LoRA adapters that do not reuse cache. Notably, we find that our modifications enable up to $58\times$ speedup for the end-to-end adapter latency[2] and over $100\times$ time-to-first-token speedup, with benefits scaling by model size and sequence length.

- Lastly, we do a deep dive into the timings of various stages in the generation process, providing insight into how the speedups are achieved.

## 2 Background and Related Work

### 2.1 Attention Mechanism and the KV-Cache

A core component of modern LLMs is the causal softmax attention mechanism, which allows models to allocate different amounts of their "attention" to different parts of the input text.[3] This mechanism consists of three components under the hood: query, key, and value vectors for each input token, which together make up the three matrices $Q$, $K$, and $V$. While each step of text generation requires only using the last row of $Q$, the entirety of $K$ and $V$ are needed in the computation. Since decoding is autoregressive, meaning that each step of the process can only use information from steps prior to it, the $K$ and $V$ for each token does not change as new tokens are processed. Therefore, these $K$ and $V$ values can be *cached*, allowing generation of tokens beyond the first to essentially only forward pass the last generated token and interact with the KV-cache in the attention mechanism.

---

[1]To our knowledge.

[2]Specifically, total time to generate 16 tokens.

[3]We omit alternative attention mechanisms such as Mamba (Dao & Gu, 2024) in the present work due to the added complexity of prefix cache reuse in those regimes.

## 2.2 LoRA Adapters

One of the most widely used PEFT methods, LoRA (Hu et al., 2021) involves training a separate matrix for updates instead of directly updating the pretrained model weights. During training, only the low-rank adapter matrices are learned while the base model weights remain frozen. During inference, these adapter matrices are added to the base model weights to achieve some desired model behavior. The adapter matrices are the product of two low-rank matrices $B$ and $A$ with dimensions $(M \times r)$ and $(r \times N)$ respectively, where $(M \times N)$ are the dimensions of the original weight matrix $W_0$ and $r \ll \min(M, N)$. When multiplied together and added to the original weight matrix, $(W_0 + BA)$ gives the weight matrix of the finetuned model.

Returning to the $Q$, $K$, and $V$ relevant to the attention mechanism and the KV-cache, for $X$ the input to the attention layer, $W^Q$, $W^K$, and $W^V$ the base model weight matrices and $\Delta^Q$, $\Delta^K$, and $\Delta^V$ the adapter matrices for the $Q$, $K$, and $V$ modules respectively, LoRA has

$$Q = X(W^Q + \Delta^Q), \quad K = X(W^K + \Delta^K), \quad V = X(W^V + \Delta^V).$$

Note that LoRA can also be applied to any linear layer in the transformer, e.g., the MLP layers.

As a PEFT method, using LoRA instead of full fine-tuning (tuning the full weight matrices directly) on large models brings enormous savings in terms of computational power and training time due to far fewer weights being updated. In terms of performance, LoRA in particular achieves high-quality results, comparable to full-parameter fine-tuning, on many tasks such as sequence classification and instruction tuning (Ghosh et al., 2024). Many variants of LoRA have been introduced, such as QLoRA, which finetunes quantized pretrained language models to achieve additional memory savings (Dettmers et al., 2023).

## 2.3 Multi-Turn Inference and Activated LoRA

AI agents—capable of harnessing multiple capabilities such as planning, automation, and reasoning to solve complex tasks—are at the latest frontier, improving the gap between artificial intelligence and human cognitive processes (OpenAI et al., 2024; de Lamo Castrillo et al., 2025). At the core of agentic thinking is the ability to reason through problem-solving in diverse real-world environments (Wang et al., 2023; Chen et al., 2025; Jin et al., 2025), made possible by multi-round interactions that break down larger goals into smaller tasks which are then solvable by reasoning or tool-use (Zeng et al., 2025; Feng et al., 2025; Yao et al., 2023; Song et al., 2023). These multi-turn interactions can be entirely internal, or they can involve human-agent collaboration as the focus increasingly shifts from human replacement to capability augmentation with AI (Zhou et al., 2025).

Within multi-round interactions, model adapters allow for agents to rotate between multiple "personalities" (Yu et al., 2024), preserve agent and user identities across conversation rounds (Wang et al., 2024), leverage additional steps in the reasoning process such as uncertainty quantification (Danilevsky et al., 2025), hallucination detection (Arteaga et al., 2024; Jia et al., 2025), prompt rewriting (Dwivedi & Mishra, 2025), and more.

Many of these adapted models have highly specialized abilities that are only needed for certain steps in a longer process. As discussed in (Greenewald et al., 2025), agentic pipelines in particular combine long contexts with specialized tasks such as control flow decisions, judging, rewriting, etc., along with traditional generalist generation tasks. These specialized tasks are prime candidates to be improved and made more robust via finetuning, but (a) full finetuning on all tasks simultaneously is challenging and expensive, and (b) LoRA models finetune well but do not allow for KV-cache reuse between adapters and the base model, slowing down inference. Indeed, every adapter switch in a pipeline would require recomputation of the keys and values for many or all tokens in the context.

Activated LoRA (aLoRA) (Greenewald et al., 2025) addresses this issue by adapting $Q$, $K$, and $V$ only for tokens occurring after the point from which the adapter is turned on via a custom invocation token sequence specified when training the adapter. The attention weight matrices of the base and finetuned models are the same up until activation of the aLoRA adapter. This enables the finetuned model to reuse elements of the base model's KV-cache instead of having to prefill all weights up until that point. Future generation using the aLoRA generates adapted $Q$, $K$, and $V$ matrices that differ from what the base model would have

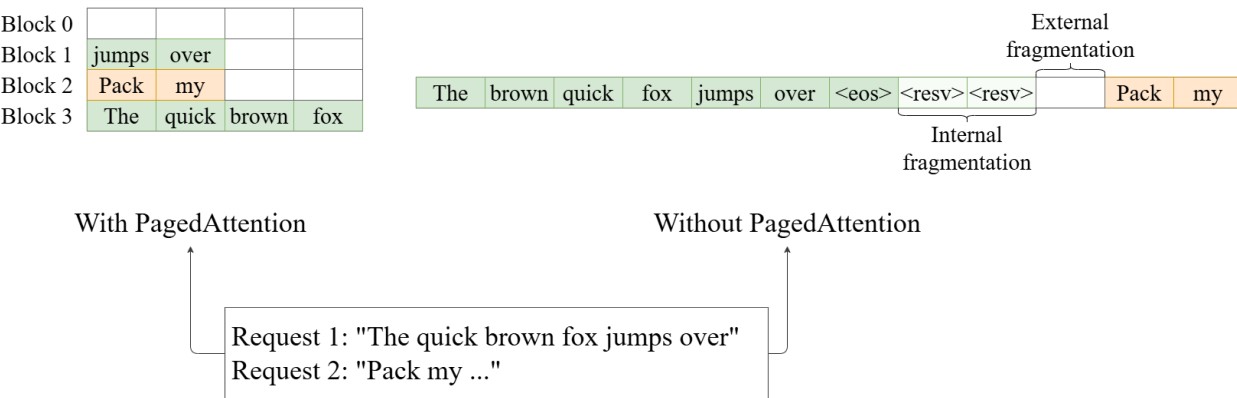

Figure 1: Physical KV-cache memory management with and without PagedAttention, adapted from Kwon et al. (2023). Example shows cache state of two requests at the same time, where <resv> represents slots reserved based on the potential maximum sequence length of a request.

generated, so eventual resumption by the base model would require prefilling its KV-cache starting from where the aLoRA had taken over. Because these adapters are trained in a way that does not alter weights for tokens occurring before the invocation sequence, a larger rank is required than that of standard LoRA adapters for the same tasks, though this has been shown to have negligible effect on inference time due to the small size of these adapters.

Returning to the attention layer, we now have

$$Q = \begin{bmatrix} X_{\text{before}}W^Q \\ X_{\text{after}}(W^Q + \Delta^Q) \end{bmatrix}, K = \begin{bmatrix} X_{\text{before}}W^K \\ X_{\text{after}}(W^K + \Delta^K) \end{bmatrix}, V = \begin{bmatrix} X_{\text{before}}W^V \\ X_{\text{after}}(W^V + \Delta^V) \end{bmatrix},$$

where $X_{\text{before}}$ is the input to the layer from tokens prior to the invocation sequence, and $X_{\text{after}}$ are the layer inputs from tokens after the start of the invocation sequence. Note that the $K$ and $V$ for tokens prior to the invocation are *not adapted*, hence they are the same as the $K$ and $V$ that would have been created by the base model for those tokens. This is exactly what is required for KV-cache reuse, and this makes the KV-cache created either by the base model or pre-invocation by an aLoRA adapter reusable in principle by the base model or any other aLoRA adapter. Realizing this reuse in a modern serving engine is the main contribution of our work.

In the context of LLM serving engines, aLoRA support can bring immense latency savings and facilitate real-time toggling of well-defined operations within specialized models known as "intrinsics" (Danilevsky et al., 2025). Examples of these well-defined operations include models finetuned for uncertainty quantification, safety checking, and hallucination detection—all of which can assist in more sophisticated reasoning within multi-turn model interactions. aLoRA has been shown to maintain competitive accuracy compared to standard LoRA and only exhibits slight performance degradation for more complex tasks, such as those involving multi-token outputs.

## 2.4 vLLM Overview

In this work, we expand the state-of-the-art LLM inference engine vLLM, which is widely used and recognized for its efficiency in high-throughput server environments (Kwon et al., 2023). vLLM significantly improves serving engine memory management by allowing KV tensors to be stored in noncontiguous memory space through PagedAttention, a concept inspired by paging from operating systems. KV tensors are grouped into "blocks" that are allocated as needed by a KV-cache manager instead of provisioning based on potential maximum sequence lengths; this completely eliminates external fragmentation and significantly alleviates internal fragmentation, with memory waste limited to partially-filled blocks at the ends of each request (Figure 1). Continuous GPU DRAM is partitioned by a block engine into physical KV blocks and mapped by the KV-cache manager to the logical blocks associated with each input request to the engine.

All requests are first assigned to a set of blocks by the KV-cache manager within a centralized scheduler and then sent to execute on distributed GPU workers (Figure 2). When the workers are available, model execution begins with the prefill stage and ends with autoregressive decoding, referencing the KV-cache manager's block table at each step within the attention layers. Blocks are assigned on an as-needed basis during decoding and freed back to the memory pool after a request has completed. The entire lifecycle of a request is split into the queue, prefill, and decode stages, demarcated by the points in time at which the request was inputted, prefill begins, decode begins, and the request is completed.

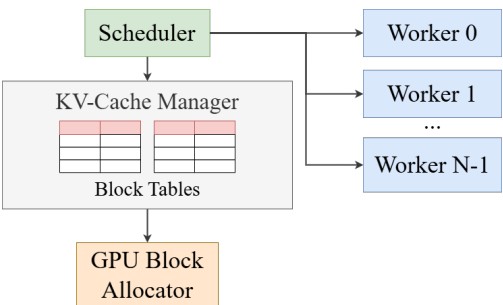

Figure 2: vLLM system overview, adapted from Kwon et al. (2023).

## 2.5 Challenges of Adapters in LLM Serving Engines

Key metrics for evaluating LLM serving engines are latency and throughput under workloads that can potentially involve thousands of adapters and requests. When LoRA adapters are invoked deep in multi-turn settings or with long prompts, the long TTFT can result in underutilized GPU resources, poor perceived system responsiveness, and potentially larger consequences in time-critical applications. For serving environments handling large quantities of requests, resource under-utilization affects end-to-end latencies of all requests due to compounding queue buildup.

Several workarounds exist to mitigate the costs incurred from the long prefill stage of LoRAs. Chunked prefill divides long prefills into multiple chunks and batches them with decode requests for better GPU resource utilization, taking advantage of the fact that prefill is compute-bound while decode is memory-bound (Agrawal et al., 2023). This lessens the impact on throughput by preventing head-of-line blocking in the engine queue. Other systems start the prefilling process on CPUs while the LoRA is loaded in on GPUs and later switch fully to GPUs for generation (Li et al., 2024). aLoRA combats this issue through the angle of prefix cache reuse, and this work is the first to analyze its performance through each stage of the serving engine inference process.

## 3 vLLM Implementation of Activated LoRA

Greenewald et al. (2025) presented an implementation of Activated LoRA based in Huggingface's PEFT library (Mangrulkar et al., 2022), now integrated fully into the PEFT package. PEFT's focus is primarily training, so its inference implementation is significantly slower than vLLM. In this work, we close this gap and fully realize modern inference speeds for aLoRA. We did this by (a) implementing cross-model prefix cache reuse as appropriate, along with (b) implementing the masking-based aLoRA architecture directly into the native forward pass within the model execution component of vLLM for use when prior prefix cache does not fully cover the history up to the invocation. We also empirically analyze speedups by serving stage, interactions with existing serving engine optimizations, and actualized performance in multi-turn, multi-adapter applications.

vLLM utilizes KV-cache block hashing for its automatic prefix caching that allows for cache reuse across requests. In LLM serving engines, automatic prefix caching primarily targets cases where multiple requests with long prompts overlap in a prefix of these prompts and can thus reuse the cache blocks of previous requests. vLLM implements automatic prefix caching by hashing each KV-cache block based on the tokens

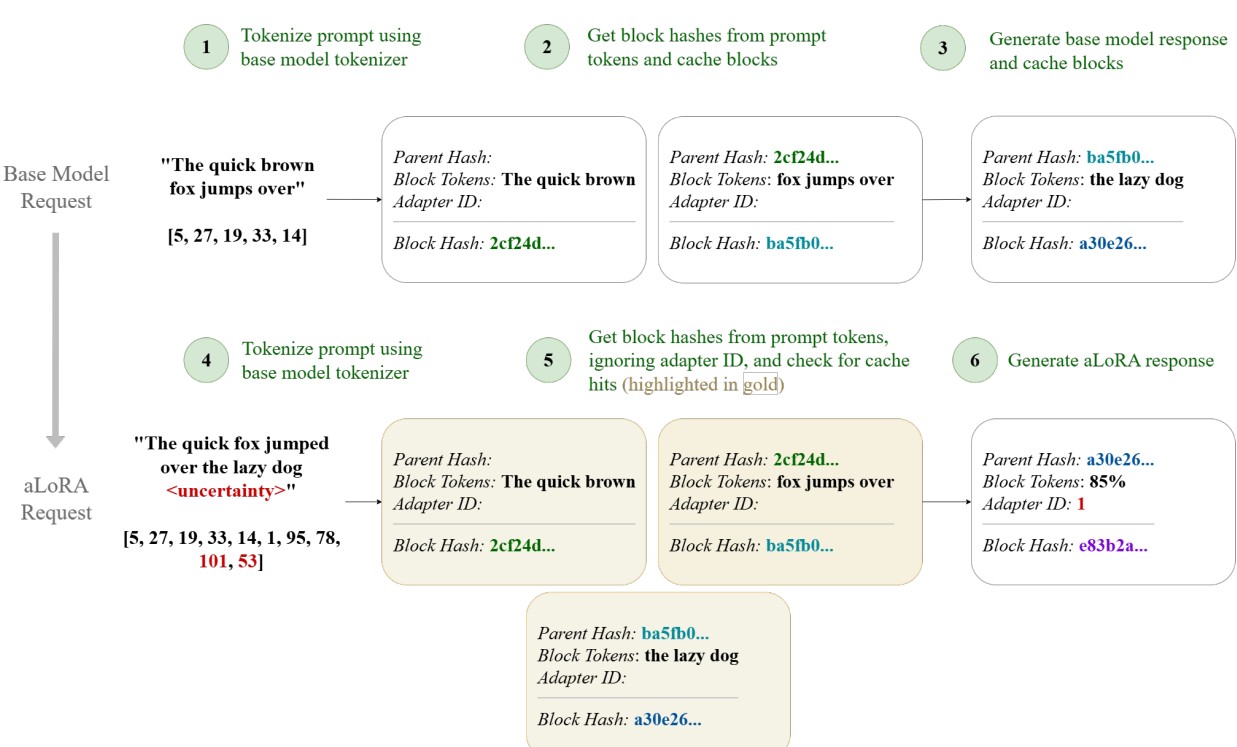

Figure 3: Example of how aLoRA reuses the base model's prefix cache within our vLLM implementation. Block size is set to 3 tokens in this example. The aLoRA activation tokens are not cached as they do not constitute a full block. Adapter ID is only incorporated into the hash of tokens generated using the adapter following adapter invocation.

within that block along with all tokens that came prior to it. This is implemented in a chaining method by hashing, for each block, (1) the tokens within that block, (2) the hash value of the blocks that came prior to it in the sequence, (3) and additional identifiers such as adapter model ID and cache salts. Although associated blocks are returned to the "free" memory pool after a request is completed, any incoming requests that require blocks whose hashes exist in the cache are able to reuse these blocks even if they are in the free memory pool. By default, KV-caches for blocks used by adapters are isolated by incorporating the internal adapter ID into the hash (whereas this field is empty for base models). By eliminating this extra step for requests that invoke aLoRAs, we allow KV-cache blocks generated by the base model to be used interchangeably with blocks from aLoRA's prefill stage (Figure 3).

This reuse goes both ways: requests using aLoRA can reuse any matching base cache blocks for their prefilling, while requests using the base model can reuse any matching blocks previously prefilled by an aLoRA request (Figure 4).

---

**Algorithm 1** Modified function where adapters are applied.

**Input:** layer $l$, input $x$, bias $b$
output $\leftarrow l.\texttt{apply}(x, b)$
base_output $\leftarrow$ clone(output)

output $\leftarrow l.\texttt{add\_adapter}(output, x, b)$
Extract mask from runtime context in forward pass.

final_output $\leftarrow$ base_output.mul(mask) + output.mul(1 − mask)
**return** final_output

---

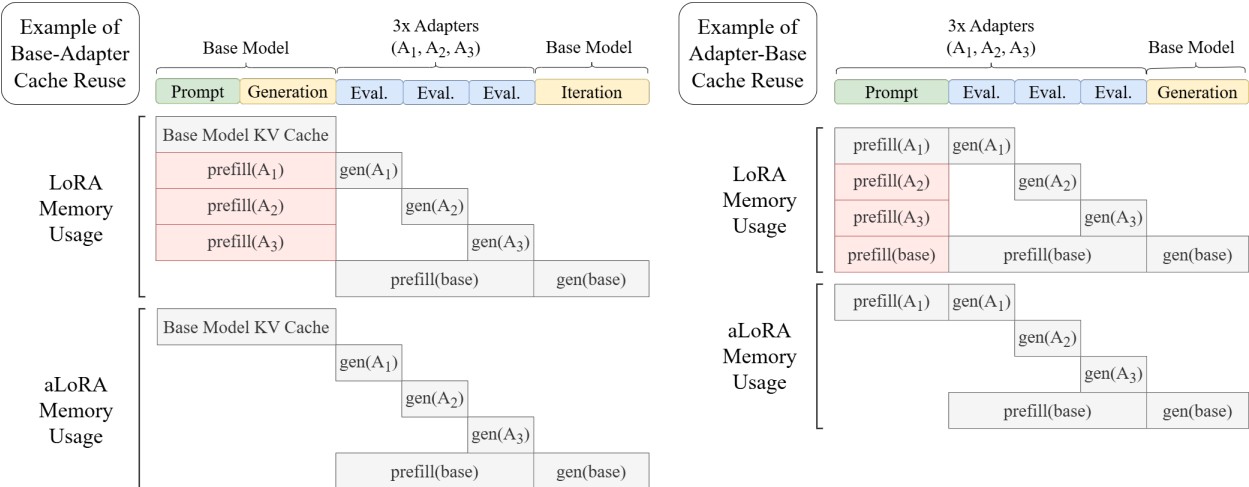

Figure 4: Comparison of LoRA vs. aLoRA cache reuse in two example multi-adapter, multi-turn pipelines. Latency savings are highlighted in red. Since input cache is interchangeable across the base model and all aLoRAs finetuned from it, latency savings scale with the number of adapters and can be leveraged in multiple settings.

Requests in vLLM's serving engine enter through one of the entrypoints (the Python library or command-line interface) before undergoing input processing, scheduling, model execution, and finally output processing (Kwon et al., 2023). With our implementation, aLoRA input requests are first identified through the existence of an "invocation tokens" field in their adapter configuration file before the location of this activation sequence in the prompt is recorded (Figure 5). Prior to each forward pass of the model, the GPU model runner prepares a piece of metadata holding information on the location (if any) of the aLoRA activation sequence within each request in the currently running batch. In order to preserve PyTorch's computational graph, we implement this information as a static tensor mask isolating tokens prior to the activation sequence and pass this down through the context manager of a model's forward pass, ensuring seamless integration with all current and future models that vLLM supports (see Appendix A). When a model is loaded into the serving engine, new QKV projection layers are used instead, which add adapter modifications only to output tokens following the start of intrinsic activation. Within a batch, the point of intrinsic activation may vary from request to request. Our implementation accommodates existing batching logic (see Appendix B), can be used in tandem with standard LoRAs, and is compatible with existing optimizations such as automatic prefix caching, chunked prefill, and torch.compile.

## 4   Evaluation

From the analysis of activated LoRA in Greenewald et al. (2025), we expect (and achieve) significant latency savings for the prefill step of the LLM inference process. In practice, we find that end-to-end, time-to-first-token, and inter-token-latency improvements are not limited to prefill improvements only but also appear throughout the serving engine inference process (queue, prefill, decode). Our evaluation in this section is thus broken down by serving engine stage to provide a more comprehensive, grounded view of the performance benefits.

### 4.1   Experimental Setup

We evaluate our modified serving engine on a set of multi-turn and multi-adapter pipelines, varying several parameters to evaluate serving efficiency under different workloads.

**Pipelines:** The structures of our test pipelines follow a simple atomic multi-turn pattern that would be the basis of longer multi-turn agentic pipelines. First, query base model $M_1$ with prompt $x$ to get response $y$.

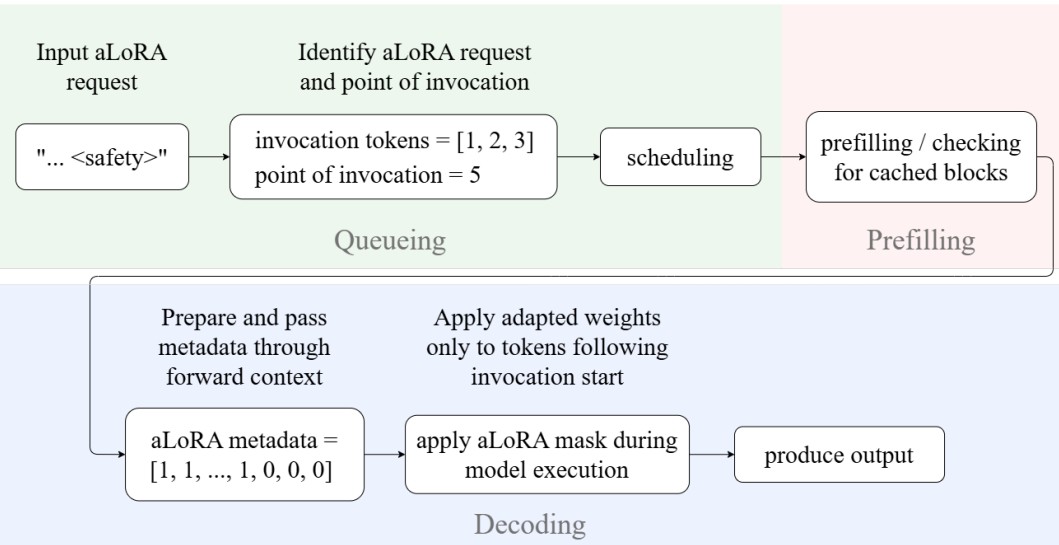

Figure 5: Life cycle of an aLoRA request from initial input to final output. This figure shows one request for simplicity, but the actual aLoRA mask covers all requests in a batch simultaneously and accounts for varying points of invocation.

Table 1: Model and server configurations.

| Model | Granite 3.2 8B | Llama 3.3 70B | Mistral Large 2 |
|---|---|---|---|
| # Parameters | 8B | 70B | 123B |
| GPUs Used | 1xH100 | 4xH100 | 8xH100 |
| Total GPU Memory | 80GB | 320GB | 640GB |
| Max # KV-cache tokens | 351,104 | 407,984 | 912,688 |

Then query adapter model $A_1$ (potentially multiple adapters) with prompt $(x + y)$ to get response $r$. Finally, in some of the trials, we feed $(x + y + r)$ back into the original base model $M_1$ to evaluate KV-cache reuse for longer conversations. This sequence encapsulates a major use case of multi-turn, multi-adapter pipelines in which model adapters are used for specialized tasks such as uncertainty quantification or jailbreak detection (Danilevsky et al., 2025) to evaluate a base model's generated response. Future turns may involve passing the evaluation along with the conversation history back to the base model for iterative response refinement (Figure 4). Such pipelines benefit from multi-adapter support, as this allows for parallel adapter evaluation and consolidated feedback in the final base model query.

We model this as a synchronous process and vary the length of the initial prompt $x$ and the base model generation $y$. Separately, we also model this as an asynchronous process and vary the request arrival rate $\lambda$. Prompts were generated randomly to fulfill the desired number of tokens. In summary, we evaluate the following pipelines:

- Synchronous base-adapter and adapter-base pipelines where initial prompt length is varied.

- Synchronous base-adapter pipeline where base model generation length is varied.

- Asynchronous base-adapter pipeline where request arrival rate and base model generation length are varied.

- Synchronous base-adapter-base pipeline where base model generation length is varied.

**Metrics:** For most of our analysis, we focus on end-to-end, TTFT, ITL, queue/prefill/decode-time, and cache hits (see Table 2 for definitions) for only the evaluation step of the pipeline, as this is where we expect to see performance differences using our modifications with aLoRA. Key metrics were collected using vLLM's Prometheus metrics endpoint, and performance was benchmarked against LoRA using unmodified vLLM.

Table 2: Key metrics and definitions.

| Metric | Definition |
|---|---|
| End-to-End Latency (E2E) | Sum of queue, prefill, and decode time for a request. |
| Queue Time | Time from request input to start of model execution. |
| Prefill Time | Time from start of model execution to start of generation. |
| Decode Time | Time from start of generation to completion of request. |
| Time-to-first-token (TTFT) | Sum of queue and prefill time for a request. |
| Inter-token Latency (ITL) | Decode time divided by # of output tokens generated minus one. |
| Cache Hit Rate | Hit rate of serving engine KV-cache across requests. |
| Throughput | Total number of tokens processed divided by E2E latency. |

**Adapters:** For these experiments, all low-rank adapters and all inputs were generated randomly, as the values of these do not affect inference speed. Pipelines were run on separate server instances for each variation of the parameters of interest. As in Greenewald et al. (2025), adapter ranks were 8 and 32 for LoRAs and aLoRAs, respectively.

**Models:** Each pipeline variant was evaluated with Granite 3.2 8B, Llama 3.3 70B, and Mistral Large 2, with models chosen to span a broad range of sizes and required computational resources (Table 1). Larger models were distributed using tensor-parallelism at server startup.

All experiments were conducted on NVIDIA H100 80GB HBM3 GPUs using CUDA 12.1, PyTorch 2.8, vLLM 0.1.dev9582, and bfloat16 precision for model weights and activations.

## 4.2 Varying Prompt Length

To emulate prompts that grow increasingly long with each round of a multi-turn conversation, we experiment with varying initial prompt length in the base-adapter pipeline described in Section 4.1. Using our modified serving engine, we evaluate performance in the setting where prompt length is varied, base model generation length is 256 tokens, and adapter evaluation length is 16 tokens. To maximize GPU utilization, batch size is chosen by dividing the total KV-cache size in tokens by the maximum sequence length of each set of trials, which includes initial prompt, base generation, adapter evaluation, end-of-sequence indicators, and aLoRA activation sequences (used in both aLoRA and LoRA trials for fairness). To ensure a fair comparison of latency changes, we fix batch size based on the largest prompt length across the set of trials with varying prompt lengths; A varying batch size that fully fills up the KV-cache for each prompt length leads to misleading trends as latency for shorter prompt lengths becomes dominated by decode times due to larger batch sizes (see Figure 15 in the appendix).

As demonstrated in Figure 6, our serving engine with aLoRA has a speedup factor over regular LoRA that scales with prompt length and model size up to 58× end-to-end and over 100× TTFT (defined as the sum of queue time and prefill time; see Figure 12 in the appendix) with a prompt length of 65k. The results shown are for the adapter evaluation step of the pipeline. In line with previous work, greater prompt lengths bring greater prefill savings (up to 45× faster than LoRA) due to increased prefix cache reuse. For example, prefix cache hit rate for adapter evaluation in a base-adapter pipeline with prompt length 1024 is 84% for aLoRA and 0% for LoRA. Note that the hit rate depends on the input length to the adapter.

We further find that serving aLoRAs over LoRAs brings meaningful speedups across all three stages, including queue, prefill, and decode time. Decode time—which constitutes the longest stage of the serving engine inference process for shorter prompt lengths—is up to 21× faster with our modifications, concentrated at

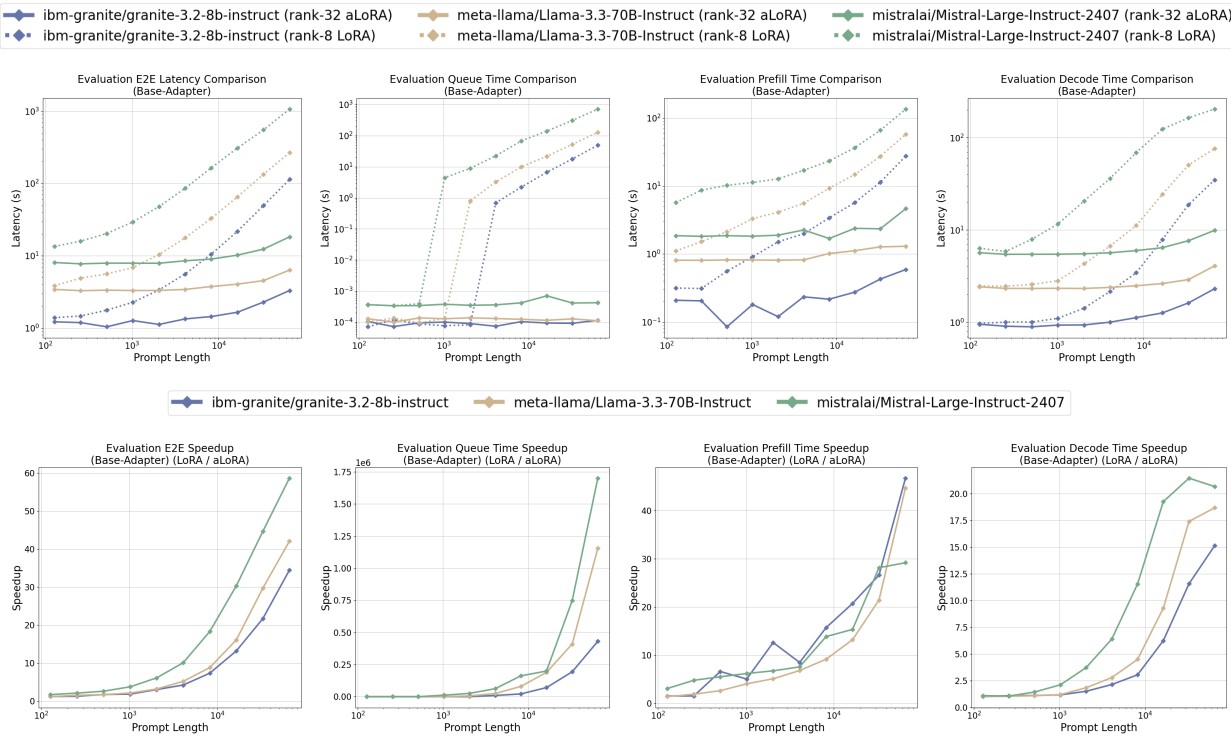

Figure 6: Comparison of serving engine inference stage latencies when using our modified aLoRA engine vs. LoRA with vanilla vLLM in a simple base-adapter pipeline. Evaluation speedups scale with prompt length and model size up to 58× end-to-end (**first column**). Queue times remain stable even with very long prompts (**second column**), prefill speedups scale up to 45× (**third column**), and we see significant decode time speedups concentrated at prompt lengths greater than 1024 (**fourth column**).

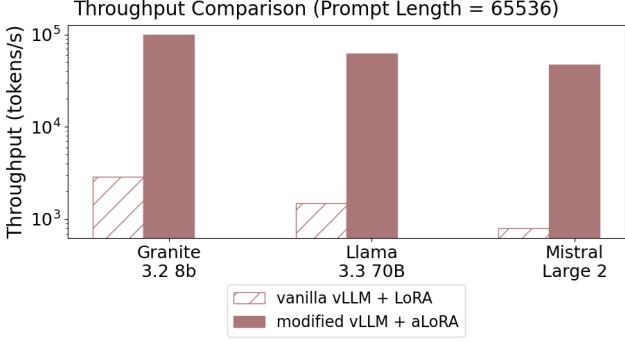

Figure 7: Token-level throughput comparison of the evaluation step in the base-adapter pipeline for LoRA vs. aLoRA with prompt length of 65k and batch size chosen to fill KV-cache.

prompt lengths greater than 1024. Increased KV-cache reuse for the prefill step decreases the number of new PagedAttention block allocations that are needed, in turn decreasing decode time for shorter prompt lengths by increasing the GPU cache hit rate. Faster decode time is likely also the main driver behind why overall speedups are greater for larger models with long prompt lengths: inter-token latency tends to increase with model size as each forward pass becomes more expensive, but we suspect that prefix cache reuse in the prefill phase improves page-level contiguity and memory access patterns in the decode phase, thus improving attention kernel streaming efficiency.

### 4.2.1 Avoiding Queue Buildup

For prompt lengths as short as 512 tokens (exact value depends on model, total sequence length, and KV-cache size), queue times spike for LoRA requests while staying constant for aLoRA (Figure 6). The long prefills incurred by requests using LoRA are broken down by chunked prefill (Agrawal et al., 2023) within vLLM to improve throughput by interleaving compute-heavy prefill with memory-intensive decode requests, however throughput and queue times are still affected nonetheless because the prefill must still be conducted. Because aLoRA skips this lengthy prefill by reusing the base model's KV-cache, our modifications enable consistently low queue times, with end-to-end latency mainly increasing from longer decode times for long prompts.

Two-way KV-cache reuse enabled by aLoRA results in similar latency savings for queue, prefill, and decode times within an adapter-base pipeline (see Figure 11 in the appendix). An example of where such a pipeline may emerge is if adapters are deployed to evaluate a prompt before it is sent to a base model for response generation. The prefill blocks generated by an adapter are able to be reused by the base model or by any other adapter finetuned from the same base model.

### 4.3 Asynchronous Latency Analysis

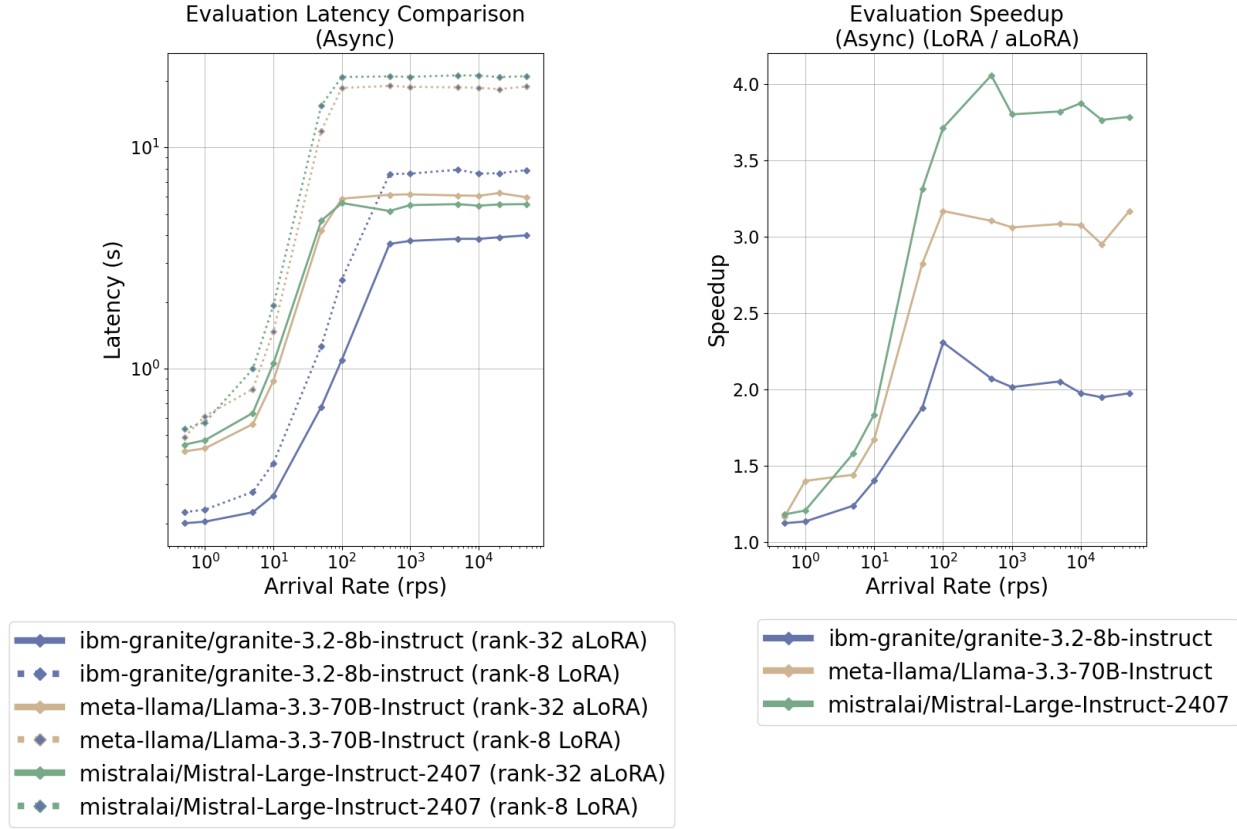

Figure 8: Latency comparison of the evaluation step in the base-adapter pipeline for LoRA vs. aLoRA in an asynchronous process. Higher arrival rates yield greater end-to-end speedups, primarily driven by queue and decode time speedups through higher GPU utilization.

Next, we model request arrivals as an asynchronous Poisson process that more closely resembles serving engine workloads in practice. Prompt length is set to 256 tokens, base generation to 256 tokens, adapter evaluation to 16 tokens, total number of requests to 500, and the arrival rate is varied. Results shown in Figure 8 reveal that maximum speedups are achieved for larger arrival rates, though benefits plateau eventually. Prefill savings are prevalent through all arrival rates, but higher arrival rates facilitate queue time savings from a

lack of prefill request backlog, and decode time savings from higher GPU compute utilization. Note that the speedup in this case is upper-bounded by the speedup in the case of full KV-cache utilization, when GPU compute is fully saturated in our pipeline. For the maximum prompt length of 65k, this matches the speedup from the synchronous trials for sequences of equal length. Note that for shorter prompt lengths, speedups in the asynchronous trials will appear higher as they represent full cache utilization, whereas the fixed batch size of the synchronous trials leads to only partial cache utilization for all but the longest prompt length.

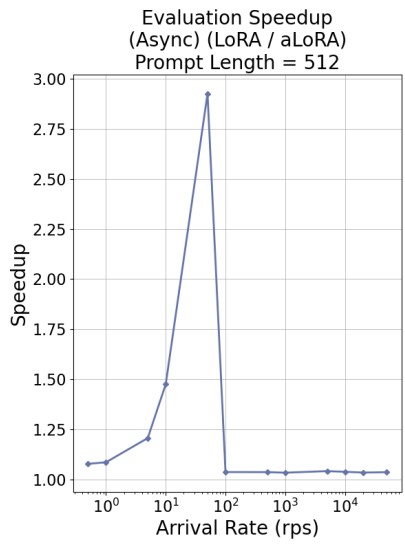
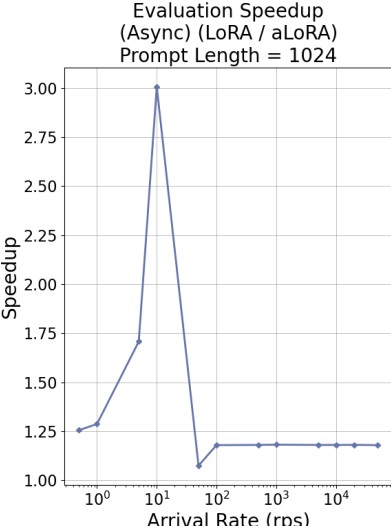

Figure 9: Speedups accelerate faster for longer prompt lengths and higher arrival rates, but care must be taken in load-balancing to avoid exceeding KV-cache capacity, as this invalidates the benefits of cache reuse.

With the number of requests set to 500, larger sequence lengths result in a drop in speedup at higher arrival rates as cache capacity is reached and previous blocks are overwritten (Figure 9), preventing cache reuse savings. Greater prompt lengths increase this peak and incur cache overflow at lower arrival rates. Our asynchronous trials suggest that with current vLLM optimizations, speedups from aLoRA's prefix cache reuse can be quite extreme but may require smart allocation of incoming requests to maximize utilization across GPUs without exceeding memory capacity. Estimation of the optimal supported arrival rate at which aLoRA speedups are still attained depends on total KV-cache size, sequence length, and an estimate of the number of requests in-flight at a given moment.

## 4.4 Varying Generation Length

When varying the length of the base model's response, we observe the same speedups as when varying prompt length because prefix caching of the first base model call does not differentiate between prefill and generated blocks (Figure 10). The savings by serving engine inference stage are also the same, with lower latency across the board for queue, prefill, and decode time of the adapter evaluation step in the base-adapter pipeline.

### 4.4.1 Multi-Turn, Multi-Adapter Pipelines

We extend the pipeline to include five randomly generated adapters—invoked in parallel—and a final, consolidated base model request with the initial prompt, generation, and adapter evaluations as input. Initial prompt length is 256, generation is 256, and adapter evaluations are each 16 tokens.

With parallel adapters, the speedup of the evaluation step remains the same because each aLoRA adapter saves the same amount of prefill over each LoRA adapter. These savings would scale with the number of adapters if they were invoked sequentially, either consecutively or in a longer multi-turn conversation consisting of repeated switches between base and adapted models. While a major contributor to lower overall latency is the adapter evaluation savings, we additionally observe a sharp increase in queue times for the

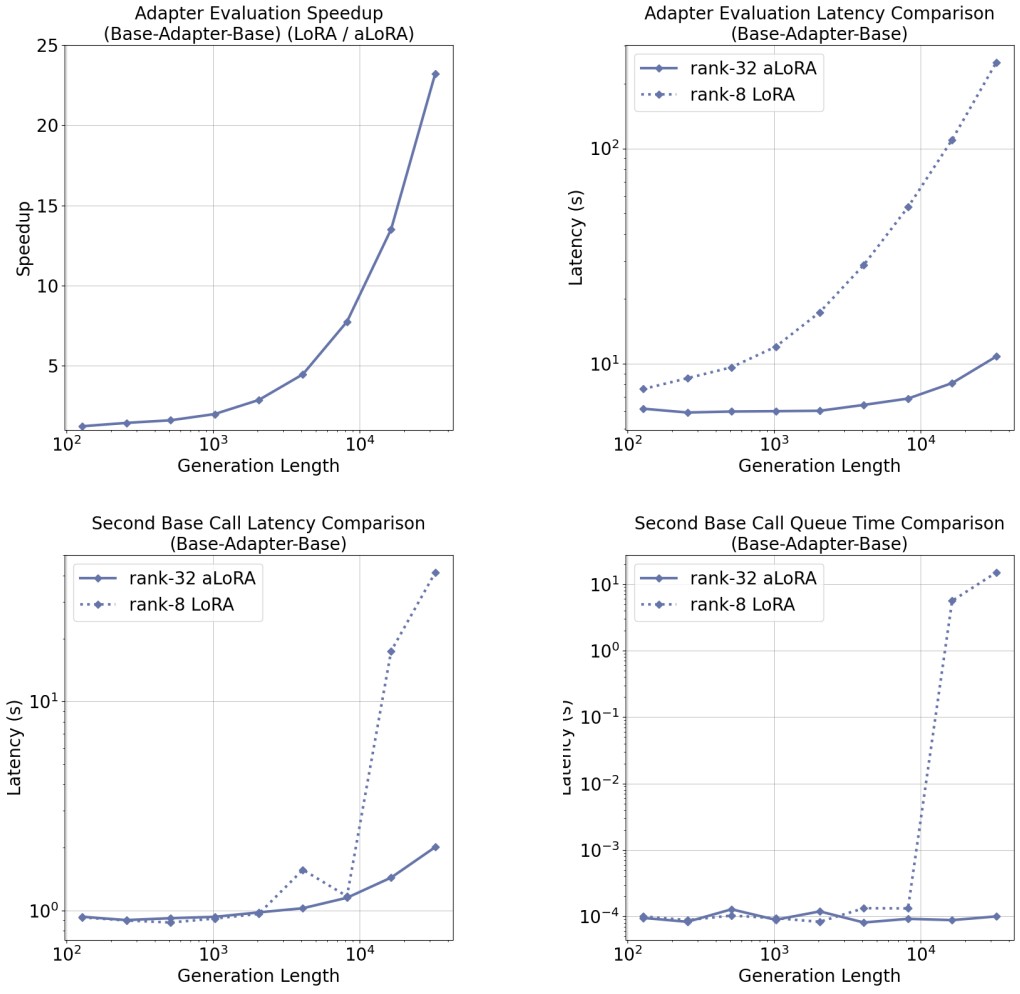

Figure 10: Latency comparison of the evaluation and second base call in the base-adapter-base pipeline as the generation length of the first base call increases. **Top row:** Speedups are the same when varying generation length vs. equivalent prompt length. **Bottom row:** Queuing delays from LoRA prefills affect TTFT of all future requests in the pipeline.

second base call of the LoRA pipeline (Figure 10). The request buildup caused by the combination of long LoRA prefills with chunked prefill affects TTFT of all future requests in the pipeline. With queuing delay worsening with each additional LoRA invoked, in addition to greater prefill and decode times from longer inputs, the advantages of serving aLoRAs scale with the number of turns and number of adapters in the pipeline.

## 5 Conclusion

In this paper, we presented the first LLM serving engine, built upon the vLLM framework, that enables efficient adapter switching in multi-turn inference through prefix cache reuse between base and adapted models via activated LoRA. We provided an implementation that seamlessly integrates cross-model prefix cache reuse with existing serving engine optimizations, fully realizing modern inference speeds for aLoRA in a set of multi-turn pipelines. Lastly, we investigated in detail how speedups are achieved in various stages of the serving engine request lifecycle, demonstrating consistent latency reductions and throughput gains across all major stages.

Future work involves enabling cross-adapter batching for improved cache reuse across heterogeneous adapter invocations, and evaluating performance on real multi-turn reasoning benchmarks that approximate agentic workflows.

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

# A  Forward Pass Integration

Pseudocode of how aLoRA metadata is passed down through forward runtime context to avoid altering model-level files.

```
# Modifications within the GPU model runner.
def execute_model(...):
    ...
    (..., alora_metadata, ...) = self._prepare_inputs(scheduler_output)
    with set_forward_context(...,
                             alora_metadata,
                             ...):
```

```
            # Execute forward pass.
            model_output = self.model(...)

def dummy_run(...):
    ...
    if lora_enabled and activated_lora_enabled:

            # Ensure that mask is captured in CUDAGraph.
            alora_metadata = build_dummy_alora_metadata()
```

## B  Mask Preparation in Batches

Pseudocode of how the activation-aware mask is prepared in coordination with vLLM's batching logic.

```
# Modifications within the GPU model runner.

def build_alora_metadata(...,
                         total_num_scheduled_tokens: int,
                         input_batch: InputBatch,
                         position_within_req: np.ndarray,
                         req_indices: np.ndarray,
                         mask1d: torch.Tensor):

    # Initialize array to be used in mask creation.
    inv_start = np.empty(shape=(num_reqs,),
                         dtype=int)

    # Extract invocation starts according to batch order.
    for request in batch:
        if adapter information is cached:
            request_index = batch.id_to_idx[request.id]
            inv_start[request_index] = request.inv_start
        else:
            inv_start[request_index] = len(request.prompt_tokens)

    # Prepare mask on CPU, isolating all tokens prior to aLoRA activation.
    mask1d_cpu = torch.tensor(
        position_within_req < inv_start[req_indices],
        dtype=torch.bool,
        device="cpu")

    # Prepare GPU tensor.
    mask1d = mask1d[:total_num_scheduled_tokens]

    # Move mask onto GPU.
    mask1d.copy_(mask1d_cpu, non_blocking=True)

    return ALoRAMetadata(mask1d=mask1d)
```

## C  Adapter-Base Pipelines

Figure 11 shows the timing results for the pipeline with varying initial prompt length, adapter evaluation of 256 tokens, followed by a base generation of 16 tokens. Identical speedups to the base-adapter pipeline demonstrate how base models are able to reuse adapter-prefilled blocks.

## D  Additional Breakdowns of the Base-Adapter Pipelines

TTFT is defined as the sum of queue and prefill times. Inference is defined as the sum of prefill and decode times (Figure 12). The decision of which to optimize varies by use case.

## E  Additional Breakdowns of the Asynchronous Base-Adapter Pipelines

Complete set of timing graphs for the asynchronous pipelines, broken down by key metrics and stages of the generation process (Figures 13 and 14).

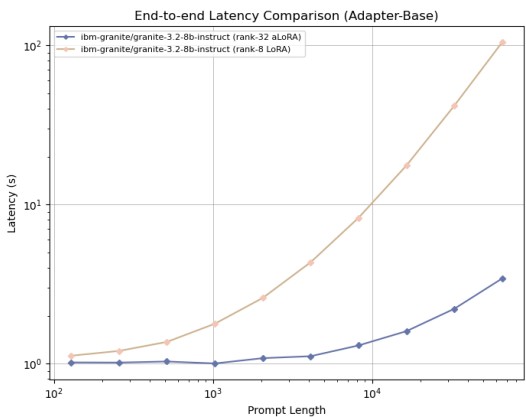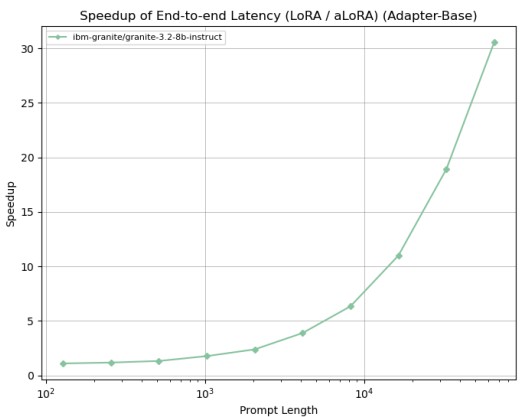

Figure 11: Latency comparison of the evaluation step in the adapter-base pipeline for LoRA vs. aLoRA.

## F  Varying Batch Size

As explained previously, we find that long decode times dominate end-to-end latency for short prompt lengths due to the greater batch size when choosing batch size to fully fill the KV-cache (Figure 15). This is why we fix the batch size in our synchronous prompt length pipelines for fair comparison of latency changes.

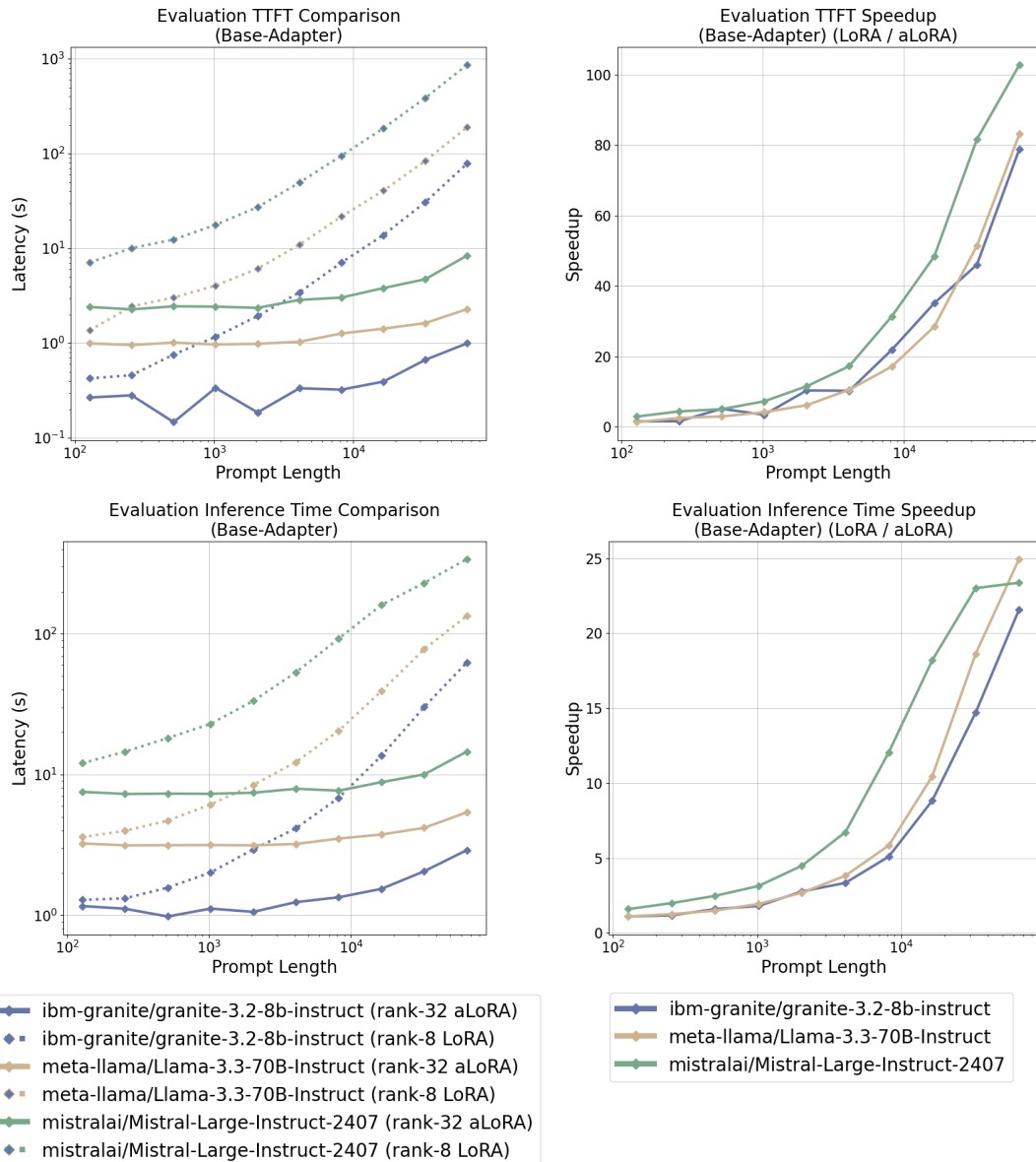

Figure 12: TTFT and inference time comparison of the evaluation step in the base-adapter pipeline for LoRA vs. aLoRA.

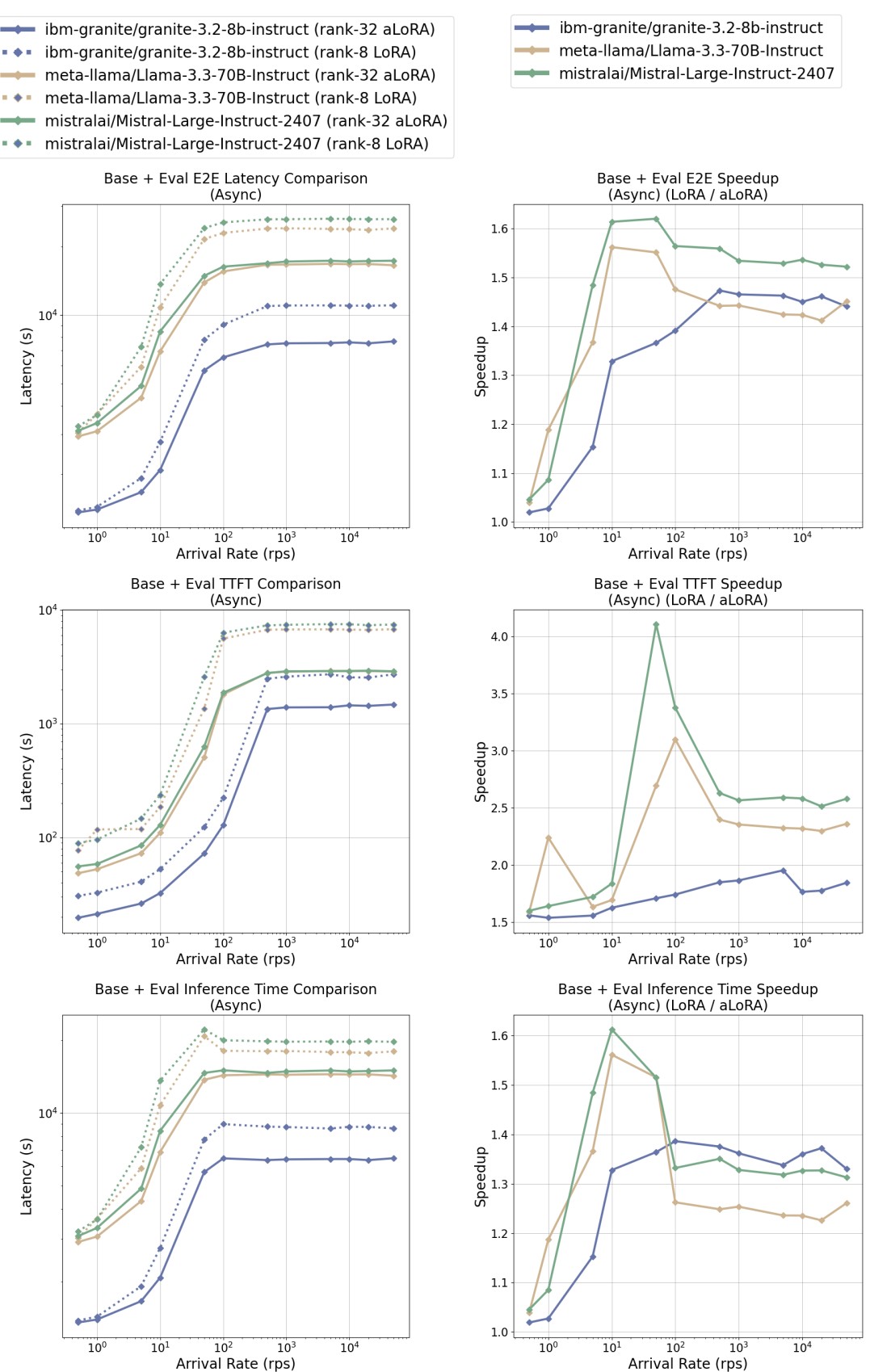

Figure 13: E2E, TTFT, and Inference (prefill + decode) time comparison of the entire base + evaluation step in the asynchronous base-adapter pipeline for LoRA vs. aLoRA.

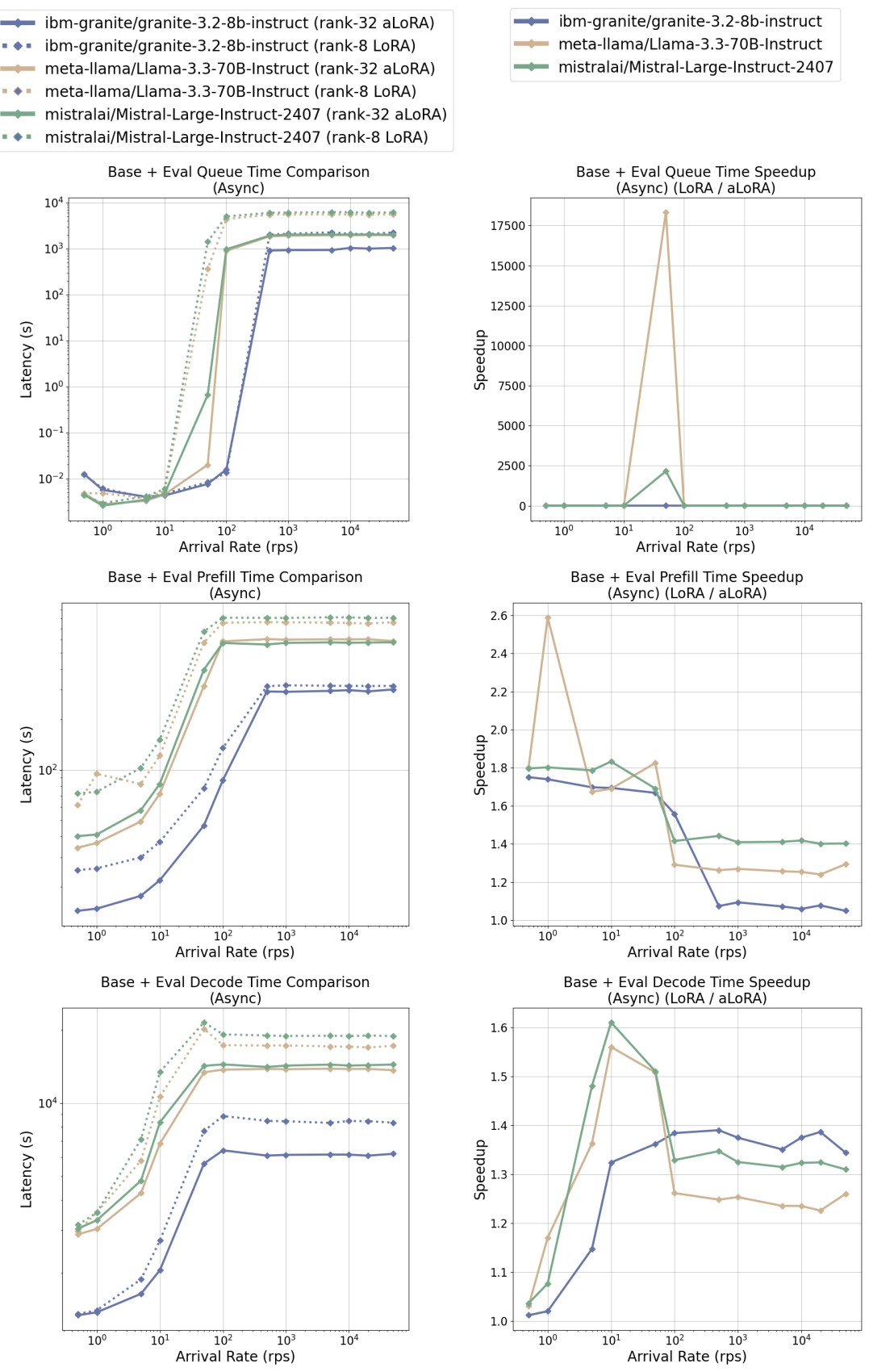

Figure 14: Queue, prefill, and decode time comparison of the entire base + evaluation step in the asynchronous base-adapter pipeline for LoRA vs. aLoRA.

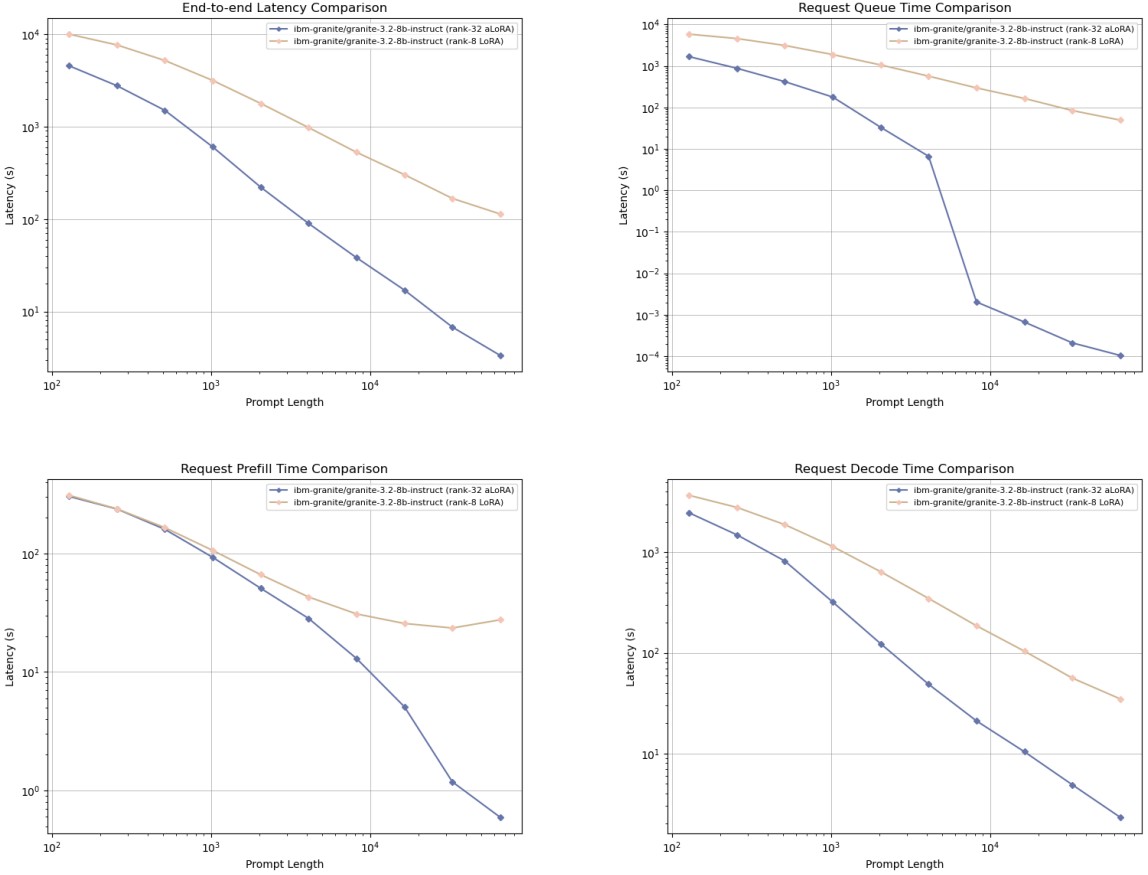

Figure 15: E2E, queue, prefill, and decode time comparison of the evaluation step in the base-adapter pipeline for LoRA vs. aLoRA.

