# OpenReview forum: "Efficient Multi-Adapter LLM Serving via Cross-Model KV-Cache Reuse with Activated LoRA"
_TMLR — Decision pending for TMLR_

### Review · Reviewer_RPcx · 2026-04-06

**Summary Of Contributions:**

This paper proposes a vLLM-based serving implementation for Activated LoRA (aLoRA) that enables prefix KV cache reuse across the base model and activated adapters, so that switching between base and adapted models does not require recomputing the entire shared prefix. The main ideas are base-aligned block hashing for cross-model caching and activation-aware masking in the forward path.

Strength:
- The inference system contribution is clear and practically meaningful,
- The implementation details are reasonably presented,
- The evaluation is categorized by queue, prefill, and decode stages.

Weakness:
- The experiments use randomly generated adapters and prompts rather than real finetuned adapters or real-world tasks, which may not truly reflect the actual performance of the proposed framework. It is noted that the paper itself leaves real multi-turn reasoning benchmarks as future work, but including at least one real case study would be very helpful.
- The paper does not directly validate the functional correctness of the modified serving path against a reference implementation, i.e., the parity check.

**Audience:**

Yes

**Audience Explanation:**

Though I am not an expert in the central topic of this paper, I expect it to be of interest to readers working on LLM serving, PEFT, and multi-turn or agentic inference pipelines. I also think the contribution is broad, extending beyond aLoRA specifically. Even readers who do not plan to use aLoRA may find the idea of enabling cache reuse across closely related models useful, especially as systems increasingly combine base models with lightweight specialization modules during inference.

**Broader Impact Concerns:**

I do not have major broader impact concerns that apply to this LLM infra work.

**Claims And Evidence:**

Yes

**Claims Explanation:**

The paper claims that cross-model KV-cache reuse with aLoRA can reduce serving latency in multi-turn, adapter-heavy pipelines and provides detailed supporting evidence. The paper also gives implementation details about each part of its design. The empirical results span multiple model sizes, workload patterns, and serving-stage metrics, rather than relying on a single benchmark.

That said, there are still some parts of the paper that the authors could improve. For example, the study is entirely about serving efficiency, so it does not actually examine whether the modified execution path exactly preserves outputs relative to a reference aLoRA implementation; neither does it test real task performance with trained adapters.

**Requested Changes:**

- Clarify the scope of the speedup numbers in experiments. The abstract highlights up to 58× end-to-end speedup and over 100× TTFT improvement; later, the paper claims that these numbers are for the adapter evaluation step. The claim should be clarified to explicitly state the gains between the evaluation step and the overall pipeline in the relevant sections.

- Add a functional correctness check for the modified serving path, as well as the parity check with the original vLLM. The paper changes both the cache hashing and the model execution path, so I would like to see evidence of end-to-end parity and correctness.

- Include at least one realistic multi-turn or agentic workload with actual prompts and trained adapters. The current synthetic setup is reasonable for latency measurement, but a real benchmark would make more sense of the practical significance.

---

> ### Author Response · Authors · 2026-06-02
> **Response to Reviewer RPcx**
>
> We thank the reviewers for their careful reading and constructive feedback. We appreciate that all three reviewers find the claims well-supported and the contribution relevant to the TMLR audience. We respond to each viewer below.
>
> **Reviewer #RPcx:**
>
> *Scope of speedup numbers:* The abstract's "up to 58x" and "over 100x TTFT" refer specifically to the adapter evaluation step of the pipeline, which is the component that benefits from cross-model cache reuse. A single "full-pipeline" speedup number would not be meaningful because overall pipeline latency depends on the base model generation length, number of adapter calls, and pipeline topology, all of which vary by application. We will revise the abstract and relevant sections to make this scoping unambiguous.
>
> *Functional correctness / parity check:* Our implementation is mathematically equivalent to standard aLoRA inference, not an approximation. There are no precision changes or approximations involved: the KV-cache stores identical vectors whether computed fresh or retrieved from cache, and our masking applies adapter weights to exactly the same tokens as a non-cached implementation. During development we routinely checked that output tokens produced by our modified vLLM exactly match those from the reference HuggingFace PEFT-based aLoRA implementation, as they must by construction. We will add an explicit statement in the paper confirming this token-level parity.
>
> *Realistic workload:* Our contribution is at the serving-engine level: the modifications affect only scheduling, caching, and masking logic without altering model weights or numerics. Because KV-cache operations are content-agnostic (latency depends on token counts, not token values), the measurements we report are representative of what trained-adapter deployments would observe at the same sequence lengths and batch sizes. As such, unless the reviewer has a specific, appropriate public benchmark in mind, any “real task” experiments would provide zero additional information beyond examples of “typical” prefix cache reuse statistics. Any attempt on our part to create such a dataset without having real customer data from multi-turn agentic applications would be entirely contrived and noninformative, and collecting such real customer data is obviously outside our ability for numerous reasons that include difficulty of collecting or approving such a dataset. We argue therefore that “real data” experiments would not help and instead note that modern agentic applications, particularly coding agents, are specifically characterized by extensive prefix reuse. This observation is consistent with recent reports on agentic coding workloads, which describe subsequent requests in long-running coding-agent sessions achieving approximately 85–97% cache hit rates after the initial cache population [1]. These findings suggest that substantial prefix reuse is a common characteristic of modern deployment workloads rather than an artifact of our evaluation methodology. We will add discussion connecting our parameterized results to realistic deployment scenarios so practitioners can reason about expected behavior for their workloads.
>
> **Summary of intended revisions:**
> * Expand Section 3 to more explicitly describe design decisions and their rationale.
> * Add subsection on cache reuse validity assumptions.
> * Specify vLLM version/commit and confirm code release upon publication.
> * Add broader impact statement.
> * Clarify that headline speedup numbers refer to the adapter evaluation step.
> * Add explicit token-level parity statement.
> * Add discussion contextualizing results for typical multi-adapter pipeline configurations.
>
> [1] “Full-Stack Optimizations for Agentic Inference | NVIDIA Dynamo Documentation,” Nvidia.com, 2026. https://docs.nvidia.com/dynamo/dev/digest/agentic-inference.

---

### Review · Reviewer_56R1 · 2026-05-07

**Summary Of Contributions:**

The paper presents a serving system for efficient multi-adapter LLM inference using cross-model KV-cache reuse with Activated LoRA. The main idea is to modify vLLM so that cache blocks generated before adapter activation can be reused across the base model and aLoRA adapters. The system introduces base-aligned block hashing and activation-aware masking in the model execution path, enabling dynamic adapter switching without recomputing the full prefix. The paper evaluates this implementation across several synthetic multi-turn and multi-adapter pipelines using models of different sizes, reporting substantial improvements in end-to-end latency, time-to-first-token, queue time, prefill time, and decode time compared with standard LoRA serving.

The main strength is the practical systems contribution. The implementation addresses a real bottleneck in adapter-based serving and is integrated into a widely used inference framework. The performance improvements are large, especially for long-context settings and larger models. The paper also provides a useful breakdown by serving stage, which helps explain where the gains originate.

The main weakness is that the evaluation is primarily based on synthetic adapters, random inputs, and simplified pipeline patterns. While this is acceptable for isolating serving latency, it leaves open questions about behavior in realistic agentic workloads, heterogeneous adapter collections, accuracy-preserving aLoRA training, and mixed production traffic. The comparison is also mainly against standard LoRA in vanilla vLLM, with limited discussion of other serving optimizations or deployment constraints.

**Audience:**

Yes

**Audience Explanation:**

This paper is likely to interest researchers and practitioners working on LLM serving, parameter-efficient adaptation, agentic pipelines, and inference optimization. Adapter-based specialization is increasingly common, and multi-turn systems often require frequent switching among specialized modules. The paper addresses a concrete serving bottleneck that arises in such settings.

The contribution is especially relevant because it connects a model adaptation technique, Activated LoRA, with practical serving infrastructure. Rather than proposing only a training method or an abstract caching idea, the paper demonstrates how cross-model cache reuse can be integrated into a production-grade inference engine. Even if some readers are not directly using aLoRA, the study provides useful insights into cache semantics, adapter switching costs, and the interaction between prefix caching and serving-stage latency.

**Broader Impact Concerns:**

The paper is primarily a systems contribution and does not directly introduce new model capabilities. The broader impact concerns are therefore limited, but several points are worth noting. More efficient adapter switching could make specialized LLM pipelines cheaper and more widely deployable, including safety checking, hallucination detection, personalization, and domain-specific assistance. This is a positive direction when used responsibly.

At the same time, lower serving cost may also facilitate large-scale deployment of specialized models in sensitive domains, including surveillance, persuasion, or automated decision support. If adapters are used for user profiling or domain-specific behavioral manipulation, efficiency improvements could amplify those risks. The paper would benefit from a short broader impact statement acknowledging that serving efficiency is application-neutral and that deployment risks depend on the adapters and pipelines being served.

**Claims And Evidence:**

Yes

**Claims Explanation:**

The central claim is that cross-model KV-cache reuse with aLoRA can substantially reduce latency when switching between base and adapted models. This claim is well supported by the implementation details and the latency measurements. The paper clearly explains why standard LoRA invalidates KV-cache reuse and how aLoRA permits cache sharing before the activation sequence. The modifications to vLLM, including base-aligned block hashing and activation-aware masking, are described concretely enough to make the systems mechanism plausible.

The empirical evidence is also convincing for the paper’s main systems claim. The authors evaluate multiple model sizes, including 8B, 70B, and 123B models, and report results across synchronous and asynchronous settings, varying prompt length, generation length, and request arrival rate. The breakdown into queue, prefill, decode, TTFT, and end-to-end latency is useful and supports the claim that benefits are not limited only to prefill savings.

However, some broader claims should be interpreted more cautiously. The evaluation does not use real trained task adapters or realistic downstream agent benchmarks, and the adapters and inputs are randomly generated. This is reasonable for measuring serving overhead, but it does not fully establish the practical end-to-end benefit in real multi-adapter applications. The paper would be stronger if it included at least one realistic pipeline with trained aLoRA adapters and task-level quality checks.

**Requested Changes:**

1. Add at least one realistic end-to-end workload using trained aLoRA adapters rather than only random adapters and random inputs. The current experiments are strong for isolating latency, but they do not demonstrate that the proposed system preserves task utility in realistic multi-adapter pipelines.
2. Clarify the assumptions under which cache reuse is valid. In particular, the paper should more explicitly state how activation tokens, adapter configuration, tokenizer consistency, positional encoding, and adapter placement affect correctness of cross-model cache reuse.
3. Provide more implementation and reproducibility details. The paper should specify whether the code will be released, which vLLM commit or branch is used, how much of the implementation is general across model families, and what limitations exist for different attention implementations or LoRA target modules.

---

> ### Author Response · Authors · 2026-06-02
> **Response to Reviewer 56R1**
>
> We thank the reviewers for their careful reading and constructive feedback. We appreciate that all three reviewers find the claims well-supported and the contribution relevant to the TMLR audience. We respond to each viewer below.
>
> **Reviewer #56R1:**
>
> *Realistic end-to-end workload with trained adapters:* As discussed in our response to Reviewer RPcx, our serving-engine modifications are content-agnostic and do not alter model weights or numerics, so our latency measurements generalize directly to trained-adapter deployments. As such, unless the reviewer has a specific, appropriate public benchmark in mind, any “real task” experiments would provide zero additional information beyond examples of “typical” prefix cache reuse statistics. Any attempt on our part to create such a dataset without having real customer data from multi-turn agentic applications would be entirely contrived and noninformative, and collecting such real customer data is obviously outside our ability for numerous reasons that include difficulty of collecting or approving such a dataset. We argue therefore that “real data” experiments would not help and instead note that modern agentic applications, particularly coding agents, are specifically characterized by extensive prefix reuse. This observation is consistent with recent reports on agentic coding workloads, which describe subsequent requests in long-running coding-agent sessions achieving approximately 85–97% cache hit rates after the initial cache population [1]. These findings suggest that substantial prefix reuse is a common characteristic of modern deployment workloads rather than an artifact of our evaluation methodology. We will add discussion mapping our parameterized results to concrete deployment scenarios, continuing to make it abundantly clear that typical agentic LLM use cases have vast KV cache reuse opportunities.
>
> *Assumptions for cache reuse validity:* We will add a subsection enumerating conditions for valid reuse, in particular that adapter placement is limited to attention QKV projections (so that hidden states and thus KV values remain identical pre-activation). We will also clarify that non-attention LoRA targets are not currently supported.
>
> *Implementation and reproducibility:* We will specify the vLLM version/commit, supported architectures, portability to other engines, and commit to open-sourcing the implementation upon publication.
>
> *Broader impact:* We will include a short statement acknowledging that serving efficiency is application-neutral and that deployment responsibility lies with operators.
>
> **Summary of intended revisions:**
> * Expand Section 3 to more explicitly describe design decisions and their rationale.
> * Add subsection on cache reuse validity assumptions.
> * Specify vLLM version/commit and confirm code release upon publication.
> * Add broader impact statement.
> * Clarify that headline speedup numbers refer to the adapter evaluation step.
> * Add explicit token-level parity statement.
> * Add discussion contextualizing results for typical multi-adapter pipeline configurations.
>
> [1] “Full-Stack Optimizations for Agentic Inference | NVIDIA Dynamo Documentation,” Nvidia.com, 2026. https://docs.nvidia.com/dynamo/dev/digest/agentic-inference.

---

### Review · Reviewer_F7sC · 2026-05-19

**Summary Of Contributions:**

The paper proposes a efficient implementation of aLoRA method, which avoids recomputing the KV cache prefilling for the adapters when using vLLM as inference engine. The implementation leads to substantial speedup in both queue time and generation throughput.

**Audience:**

Yes

**Audience Explanation:**

Reducing inference latency cost is of general interest of LLM infrastructure community.

**Broader Impact Concerns:**

Not applicable.

**Claims And Evidence:**

Yes

**Claims Explanation:**

The numerical experiments show that aLoRA with the proposed implementation achieves substantial efficiency improvement: Granite 3.2 (8B) on 1× H100, Llama 3.3 (70B) on 4× H100, and Mistral Large 2 (123B) distributed over 8× H100 via tensor-parallelism. The authors also present isolated stage-by-stage latencies in Figure 6.

**Requested Changes:**

The inference acceleration over standard LoRA is a natural feature of aLoRA method; there seems to have no special design of the proposed engine. While I understand that one needs to modify the vLLM engine to support the feature, it does not require system-level design but just the implementation-level adjustment, e.g. disable the Adapter ID during the forward of base model. Hence, I suggest the authors to highlight their technical contributions in the paper.

---

> ### Author Response · Authors · 2026-06-02
> **Response to Reviewer F7sC**
>
> We thank the reviewers for their careful reading and constructive feedback. We appreciate that all three reviewers find the claims well-supported and the contribution relevant to the TMLR audience. We respond to each viewer below.
>
> **Reviewer #F7sC:**
>
> We will add more clarification on the technical contribution below and in the revision. Realizing cross-model cache reuse in vLLM required several coordinated modifications: (1) base-aligned block hashing that strips adapter identity from pre-activation blocks while preserving correctness post-activation, (2) activation-aware masking via PyTorch's context manager to apply adapter weights only beyond the invocation point, and (3) accommodating heterogeneous activation points within a single batch while remaining compatible with chunked prefill, torch.compile, and tensor parallelism. In the revision we will expand Section 3 to more explicitly describe these design decisions and their rationale.
>
> **Summary of intended revisions:**
> * Expand Section 3 to more explicitly describe design decisions and their rationale.
> * Add subsection on cache reuse validity assumptions.
> * Specify vLLM version/commit and confirm code release upon publication.
> * Add broader impact statement.
> * Clarify that headline speedup numbers refer to the adapter evaluation step.
> * Add explicit token-level parity statement.
> * Add discussion contextualizing results for typical multi-adapter pipeline configurations.

---

### Decision · Action_Editor_Xh6e · 2026-07-10

**Recommendation:** Accept as is

**Audience:**

Yes

**Audience Explanation:**

Reviewers were unanimous on this. I believe Reviewer 56R1 summed it up quite nicely. One particularly important example is that the increasing focus on things like multi-turn interactions may benefit from adapter switching. This work is also clearly useful follow-up from prior work on this subject (aLoRA), and I think is clearly of interest in the same way that the prior work was of interest. I would also like to say that making the more abstract method work on the system side is a good example of the type of contribution that often generates interest. Given how important related frameworks (eg. vLLM) have become to the research sphere, I think the argument for interest is clear.

**Claims And Evidence:**

Yes

**Claims Explanation:**

I believe that most of the substantive discussion in the reviews broadly supports the fact that the claims are substantiated by clear evidence. The one exception raised in 2 reviews to this was the lack of "realistic" workloads. I believe that the authors have done a good job discussing why this is difficult. I think that the lack of this does not impinge on the work's evidence, but instead points to difficulties in measuring realistic workloads in the space.

That being said, I believe this is worth calling out explicitly in the paper. The authors have already discussed ways in which they will attempt to make connections between their experiments and what features of "realistic" workloads likely exist, so I believe that this is fully addressed.